# Self-Participation Experiences among Well-Adapted Hemodialysis Patients

**DOI:** 10.3390/healthcare9121742

**Published:** 2021-12-17

**Authors:** Li-Yun Szu, Lee-Ing Tsao, Shu-Chuan Chen, May-Lien Ho

**Affiliations:** 1Department of Nursing, Taoyuan Chang Gung Memorial Hospital, National Taipei University of Nursing and Health Sciences, Taoyuan City 33372, Taiwan; szu4069@cgmh.org.tw or; 2School of Nursing, National Taipei University of Nursing and Health Sciences, Taipei City 112303, Taiwan; 3Shin Kong Wu Ho-Su Memorial Hospital, Taipei City 111045, Taiwan; 4Department of Hemodialysis, Shin Kong Wu Ho-Su Memorial Hospital, Taipei City 111045, Taiwan; R0002969@ms.skh.org.tw

**Keywords:** hemodialysis, self-participation, grounded theory

## Abstract

A successful self-participation experience empowers patients to adapt to living with hemodialysis. However, few studies regarding the subjective experiences of such patient participation have been conducted. This study’s purpose was to describe hemodialysis patients’ perspectives on integrating hemodialysis into a new life regarding self-participation experience. A qualitative study using the grounded theory method was applied. Thirty-two well-adaptive hemodialysis Taiwanese patients attended in-depth interviews. “Integrating hemodialysis into a new life journey” was identified as the core category guiding the entire self-participation experience of hemodialysis patients. The three antecedent themes were “Sense of worthlessness”, “Life is still worth living”, and “Friendly and joyful atmosphere of the hemodialysis room”. Once the patients went through the three antecedent themes, they gradually began making efforts to participate more fully in their hemodialysis. Within this participation experience, the hemodialysis patients exhibited these four interactive themes: “Overcoming one’s predicament”, “Integrating self-care skills into my life”, “Resuming previous roles and tasks”, and “Adapting to independent living”. Finally, most adaptive patients master the hemodialysis life. Encouraging patients to discover that their life is worth living and providing a friendly and joyful atmosphere in hemodialysis units are the keys to facilitating patients’ self-participation more fully.

## 1. Introduction

Each patient is an expert regarding his or her own body, symptoms, and conditions [1]. When patients are able to influence and participate in their own healthcare, they can view a disease from their own perspectives while considering their personal habits, potential, rights, opportunities, preferences, experiences, feelings, and fears in the decision-making process of their treatment [2,3]. Thus, the World Health Organization (WHO) believes that the development of patient empowerment, involvement, and participation is linked to the promotion of patient safety. Patient participation is defined as patient involvement in decision-making in their health care, safety management, and compliance with prescriptions [4]. Patients with end-stage chronic kidney disease (CKD) typically undergo hemodialysis three times per week, with each session requiring 3~4 h of dialysis time. The discomforts associated with hemodialysis include limitations of the patients’ fluid and food intake and physical activity, as well as pain, itching, fatigue, weakness, feelings of inadequacy, and negative perceptions. These may include role restrictions and social isolation, which impose physical, mental, and, sometimes, economic and social burdens on dialysis patients [5,6,7,8,9,10]. Consequently, hemodialysis patients are unable to undergo normal lives. Thus, we uncover how hemodialysis (HD) patients go through the negative experience and return to normal life.

Many recent studies have indicated that CKD requires long-term self-management or individual strategies for coping with stress [5,11,12]. Thus, CKD patients must reinterpret their disease, establish new life perspectives, and attempt to better adapt to the long-term living conditions associated with their disease, allowing them to enhance their self-worth [2,13,14,15]. During the decision-making process, patients encounter the “bridge” in patient participation. This bridge allows the patient to translate his or her knowledge, experiences, and individualized learning into behavior that may improve his or her quality of life, or the patient is able to completely express his or her feelings, physical, and mental states [2]. The participation process contributes to decision-making and shared self-determination or self-care into daily life, enabling the patient to develop a new life [2].

However, previous studies have focused on empowerment [2,15,16,17,18,19] and self-management [15], healthcare professional perspectives of patient participation [12,16,20,21], as well as working-age HD adults experiences of participation [10]. Thus, the present study was designed as a qualitative study that describes the perspective of well-adapted hemodialysis patients regarding their self-participation experience. The findings of this study may help healthcare professionals to understand the self-participation experiences of hemodialysis patients, and apply the findings to help unadjusted HD patients undergoing the HD trajectory become involved in their HD treatment and management.

## 2. Materials and Methods

### 2.1. Study Design

A qualitative study was conducted applying the grounded theory method that enables researchers to study a specific phenomenon or process and uncover new stories based on the collection and analysis of real-world data. The participants were recruited from an outpatient hemodialysis clinic with 126 beds at a medical center in Taiwan using purposive sampling. The sampling criteria for participants were as follows: (a) had already adapted to life as a hemodialysis patient and was leading a normal life; (b) was aged 20 years or older (c) had been undergoing hemodialysis three times a week for at least one year, and (d) they identified themselves as undergoing well-adapted HD, although some interviewed HD patients were very long HD survivors who still have a vivid memory of initial self-participation experiences and were willing to share these experiences. The well-adapted patients were referred by attending physicians or the head nurse.

### 2.2. Ethics Statement and Data Collection

The study protocol was approved by the Institutional Review Board (IRB) of a medical center in northern Taiwan (ethical approval number: 20180304R). In order to ensure informed consent and to protect the participants’ anonymity, confidentiality, and interests, the participants were provided with sufficient time to read the consent form. All participants were chosen to be interviewed once during a scheduled dialysis session in a private room at their dialysis centers, in order to save time and reserve energy, without nurses or family in the private room. Prior to beginning the interview, the purpose of the study was explained again, and the included patients provided signed informed consent. The female first researcher was a nephrology nurse practitioner. SLY interviewed each interviewee using an interview guide to conduct face-to-face interviews (Table 1). All interviews were audio-recorded.

### 2.3. Interview Content and Analysis

For the purpose of preventing presupposition bias and to ensure consistency of the interview content, open questions jointly developed by the three researchers were used in the interviews, during which the interviewees could provide vivid descriptions of their life experiences regarding HD self-participation. Open-ended interview prompts were also utilized to encourage the interviewees to provide in-depth accounts. Prompts and exploratory questions were utilized, and verbatim records were made until theoretical saturation was reached. The content of each interview was converted into an interview record within 48 h after recording. Comparative methods were used consistently to analyze the transcripts.

Verbatim transcripts of interviews and field notes were analyzed using the constant comparative method for repeatedly comparing the existed categories and content. Four steps were executed during the coding process: (1) open coding: the researcher repeatedly reviewed each interview transcript to obtain insight into the meaningful phrases and sentences by line-by-line; (2) axial coding: the related subcategories were clustered to generate main themes; (3) selective coding, which integrated the axial coders; and (4) determination of the initial diagram: all subcategories were testified to find out the core category, build up the storyline and develop an initial substantive theory about the self-participation experiences, until category saturation was achieved and no new major themes arose. Four criteria were applied to evaluate the trustworthiness of this study: (a) credibility, (b) transferability, (c) dependability, and (d) confirmability [12]. This was achieved by recruiting participants with a broad range of diverse backgrounds, a peer-review coding process, transcribed, audio-taped interview notes, and theoretical and methodological notes, to ensure the objectivity of data collection and classification.

## 3. Results

This study screened and recruited 32 HD patients, including 15 male patients and 17 female patients, with an average age of 61.72 years (Table 2) and an average HD period of 9.06 years. No participants were on the waiting list for kidney transplantation.

Initially, the HD patients often self-reported feeling that they were “worthless” individuals. When affected by these feelings, the patients experienced fear, sadness, and negative feelings arising from their need to rely on others and the discrimination that they faced. However, these patients discovered that “life is still worth living” and recognize the “friendly and joyful atmosphere of the HD room.” Consequently, these two forces would drive them to participate in their treatment more completely and gradually integrate hemodialysis into their new life journeys. Six themes emerged during this process (Figure 1).

### 3.1. Antecedent Theme: Sense of Worthlessness Being Gradually Disintegrated

#### Perceived Survival Value and Supportive Hemodialysis Room

Self-perceived survival value and a “friendly and pleasant atmosphere in the HD room” initiated patients’ self-participation. Upon learning that they required HD, patients experienced negative feelings arising from their need to depend on others and the discrimination that they would face. In some cases, patients may even discontinue the treatment or become pessimistic. People who felt a “Sense of worthlessness” indicated that they experienced numerous negative feelings because HD patients are subjected to several restrictions in their daily lives and must rely on a machine and the hospital for the remainder of their days. In some cases, patients may even be disparaged by their relatives, friends, and neighbors, or experience a sense of worthlessness due to their disability or the fear that they will become a burden to their family and friends and be incapable of living independently.

*“The doctor told me that I had to go on hemodialysis, and I refused. I insisted on not going and ran away from the problem. …A man who is told that he needs hemodialysis feels worthless. My wife was always crying…”*.(P10)

*“When I knew that I had to undergo HD, I was in pain and fearful. My girlfriend kept asking about the situation that we couldn’t get married, work, or raise our children. With tears in her eyes, she told me that it was alright because I wouldn’t die immediately. I am able to live a long life while I’m on HD. For the sake of my marriage and children, I decided to do it…”*.(P07)

*“I cried from the pain of the needle insertion. I had to restrict my daily food intake, and I had no freedom. I asked them how they could be happy. This was when I learned about the many types of local anesthetics”*.(P11)

*“Initially, I didn’t know anyone when I started my hemodialysis. I listened to what the others (fellow dialysis patients) were talking about and whether they had the same symptoms as I did. I figured that I should follow what they were doing, and it’ll be effective…”*.(P05)

### 3.2. Interactive Themes

#### 3.2.1. Overcoming One’s Predicament

The HD patients were brave enough to confront the tough situation by relying on their religion or beliefs; they engaged in self-talk to endure the distress caused by HD needle insertions, forced themselves to achieve better results through various methods, and thereby adapted to the discomfort of HD through positive feedback from self-encouragement.

*“As a devout Christian, I felt that God was punishing me. However, in my prayers, He told me that He has his reasons, and I should accept this ordeal. Every time I had a needle inserted, I thought about God sending his angels to lift away my pain. I attempted the methods of the others (fellow HD patients). I felt myself improving… because these methods were helpful”*.(P04)

#### 3.2.2. Integrating Self-Care Skills into My Life

The patients’ integration of self-care skills into their lives occurred during the self-participation experience. Hemodialysis imposes various life restrictions, including those related to electrolyte balance (diet) and fluid management. After four hours of HD, the arm in which the HD needle was inserted will be mostly immobilized and cannot be turned; hence, only one arm is freely movable. Furthermore, the patients must gradually acquire HD skills. Based on these circumstances, the patients needed to learn the following self-care life skills: water restrictions, fatigue management, traveling to the HD center by oneself, self-preparation of a homemade diet to maintain good energy and nutrition, self-relief of discomforts, maintaining AV-Shunt function patency and staying alert for changes in weight and laboratory measurements.

*“I was unable to restrict my water intake…In summer, I would suck and swallow a lot of ice…and I swallowed more than I spit out. Then, someone taught me how to make iced lemon water, which seemed to help cope with my water restrictions. I would place lemon water in a bottle, freeze it and drink after exercising. It was effective, as I would drink the amount of ice that melted…”*.(P24)

*“They (fellow dialysis patients) taught me that I could drink more. I may not drink four hours prior to hemodialysis, but I could drink more during each session and during meals. I thought to myself that I could eat more if I didn’t increase my weight; I could eat less if my blood potassium was not controlled. I ensured that vegetables were cooked. When I drank water during each session, I wouldn’t exceed my restrictions or have low blood pressure. I didn’t feel weak, as I had enough nutrients. Therefore, I would take a bus by myself to the hemodialysis room and arrange a taxi by myself to go home”*.(P01)

*“Blood vessel care is very important. I am afraid of the occlusion of the AV shunt at night, and I don’t even dare to sleep. It would result in me being unable to “wash” my kidneys (dialysis). I want to use an L-shaped pillow to prevent occlusion. When I wake up, I will touch the blood vessel. If the flow is not strong, I will strengthen my grasping muscles and let the blood flow improve ”*.(P15)

#### 3.2.3. Resuming Roles and Tasks

When an HD patient has integrated HD into his or her daily life, he or she will be able to gradually assume the roles and tasks of a normal person, including holding a job, utilizing social benefits to reduce family burdens, caring for his or her family, and enjoying social activities. As a result, the HD patient will experience a reduction in stress related to post-hemodialysis discomfort and achieve a better physical and mental state.

*“I love beauty, but my face disappears after hemodialysis. Today, I will wear makeup. I will be beautiful when taking hemodialysis. As long as I look happy, others will feel the same. I look more beautiful”*.(P06)

*“Now, I am able to perform carpentry and sell my own creations. My wife no longer has to endure many hardships... I feel more like a man now that I have more self-worth and a sense of self-accomplishment”*.(P08)

#### 3.2.4. Adapting to Independent Living

The patients were required to make independent arrangements to engage in outdoor activities and fun activities, develop their own personal interests and life, bring positive energy into their lives, and help their families or others understand that they were living well.

*“I would always check with the hemodialysis centre in Tainan located nearest to my hometown before treatment. Traveling locally isn’t a problem for me, and I have been to Japan. I would undergo my hemodialysis sessions before departing”*.(P09)

*“The blood vessel on my arm is unsightly. Since I enjoy making handicrafts, I would buy fabrics and stitch to make armbands that are beautiful and special: other fellow HD patients felt the same way, so I would give armbands to them. I felt happy”*.(P20)

### 3.3. Consequence Theme: Self-Mastering Dialysis Lives

Regarding happiness, satisfaction, and appreciation, when the HD patients engaged in HD self-participation and adapted to HD through the four interactive themes of behaviors that mutually influenced each other, they began to attain joy and satisfaction in their lives after HD because they now had control, and learned to better appreciate themselves.

*“I am always able to overcome any difficulty all along..., I am grateful… I just want my family to be healthy and happy, as I am able to take care of myself. I feel happy as I take daily HD … several conversations and the government subsidies are also viewed as remunerations. You can see that we’re pretty well off, too”*.(P22)

*“I was left alone after my parents passed on. I used to be scared of the two needle insertions during each hemodialysis session because I was timid and susceptible to pain. However, now I don’t require anesthesia. I told myself to handle one thing at a time. I now have higher self-esteem since I grew from being fearful and timid to an independent person. I feel blessed and free all the time”*.(P19)

## 4. Discussion

To the best of our knowledge, this was the first study to explore the Taiwanese hemodialysis self-participation experience using the grounded theory method. This study shows that the antecedent theme of “Worthlessness sense being gradually disintegrated” contains “Perceived survival value” and “Supportive hemodialysis room” and four interactive themes: “Overcoming one’s predicament”, “Integrating self-care skills into my life”, “Resuming roles and tasks”, and “Adapting to independent living”; and finally, consequence theme of “Self-mastering dialysis lives”.

### 4.1. Antecedent Theme: Worthlessness Sense Being Gradually Disintegrated

Patients may develop a sense of worthlessness or depression-induced thoughts. Depression is a commonly encountered psychological problem among HD patients [22,23,24] and may even affect the mortality rate of the HD patient population [25,26]. Supporting the depressed patients in discovering the value of life and providing a friendly environment in the HD room will initiate patients’ self-participation in their treatment. In the present study, “friendships with others in the dialysis room” was the foremost precursor factor for self-participation. Patients were encouraged to make friends with other patients in the same HD room, and the supporting group helped them to overcome their fear, discomfort, and boredom during hemodialysis. Nurses should not only work on their HD skills but also on creating a friendly, relaxed, and light-hearted atmosphere, in which patients are praised for braving the pain of needle insertion and encouraged to overcome their feelings of exhaustion. In some cases, arrangements were even made to seat HD patients with a successful self-participation track record beside patients who were undergoing their first HD session or patients who had not adapted well to the process. By joining a group, patients were no longer fighting the disease on their own, but instead, worked alongside individuals with the same condition and with whom they shared a common treatment. This approach provides patients with space where they can talk about the suffering caused by the disease and find joy and a sense of belonging in their struggles.

Previous studies showed that adjustment to ESRD was a dynamic and constant process [27]; HD patients experienced three periods of adjustment (beginning 1–3 weeks from the first dialysis, or honeymoon period, 3–12 months of frustration period, and the long period of adjustment) [28] This process is contrary to our research. The possible reason is that our participants are all well-adapted HD patients, which confirms the advantages and importance of HD self-participation. A 2008 study proposed that the emotion-regulation strategies employed by HD patients play a significant role in their attitudes towards the disease [29,30]. The results of that study were consistent with the findings from the present study, indicating that the atmosphere of an HD room is a factor that influences the emotions of HD patients. Due to the side effects of post-hemodialysis discomfort, the various restrictions in their lives, and the frustrations that they face, HD patients were reported to need to draw strength from their religion or beliefs and strengthen their problem-solving skills and ability to adapt: the findings from those studies are consistent with those from the present study [29,30,31]. These patients utilize strategies characterized by self-encouragement and self-talk to force themselves to endure the pain of needle insertions; they also attempt several methods until they are able to integrate post-hemodialysis discomfort into their lives. Furthermore, they also incorporate self-rewards into their downshifted or slow-paced lives.

The introduction of religious or spiritual interventions for chronic HD patients helps trigger positive emotions during stress and influences overall health (measured using quality of life indices), including patients’ commitment and adherence to hemodialysis [32,33,34]. A systematic review of 33 studies examining spiritual aspects of patients with end-stage renal disease (ESRD) demonstrated a positive correlation between spirituality and the well-being of patients with ESRD. Six of those quantitative studies revealed a relationship between spirituality and the quality of life of HD patients, and generated findings similar to the data from the present study, namely, the role of religion and beliefs in the ability to overcome difficulties. Some scholars have also proposed that coping strategies characterized by problem-centered participation are linked to improvements in the survival, physical function, and mental health of hemodialysis patients [35].

### 4.2. Interactive Themes: Overcoming One’s Predicament, Integrating Self-Care Skills into My Life, Resuming Roles and Tasks, and Adapting to Independent Living

Patients undergoing HD attempt various methods and learn a number of techniques to help them adapt to hemodialysis-related problems in areas such as transportation (e.g., to the HD center), diet, fluid intake, exhaustion, blood pressure fluctuations, and cramps. These methods help patients to reduce the obstacles created by HD in their daily lives and the burden on their families, as well as to build their confidence in playing their roles and taking on tasks, obtain positive feedback relating to their efforts, and maintain their social lives. In recent years, many studies have shown that the promotion of problem-centered participation was recommended to lengthen the lifespan and improve the quality of life of HD patients [11,35].

In the present study, the theme exhibited by the self-participating HD patients emphasized the attainment of results concerning self-awareness, self-efficacy, and self-control. The patients actively participated in individualized learning to acquire the HD skills they needed to utilize social resources, regain control over their lives, and understand how they would be able to achieve and appreciate the outcomes related to self-participation in HD. Throughout this process, the HD patients continued to integrate HD into their new life journeys.

The present study found that participants expressed that as they gradually resume their roles and lives, they would regain self-confidence and adapt to an independent life. After hemodialysis, patients needed to rebuild their lifestyles to accommodate the hemodialysis schedule. In many cases, patients would travel back and forth independently from home to the HD center and needed to learn how to deal with their discomforts when they were alone before and after dialysis. Their daily activities were different from those of their peers and friends. Therefore, teaching patients to be alone and help themselves during activities, shopping or entertainment is an important step in self-participation. Because of self-participation in HD life, they would perceive the usefulness of self-care, self-efficacy, and health status in coping, compliance with fluid and food restrictions, sleep disorders, symptom distress, as well as adjustment of anxiety, emotional distress, and dialysis stress [36,37]. Participants expressed that as they gradually resumed their roles and lives, they regained self-confidence and adapted to an independent life. After HD, patients needed to rebuild their lifestyle to accommodate the HD schedule. In addition, participants expressed that through self-participation, the dialysis life has been integrated into their life, and considering the complications of kidney transplantation, such as infections and rejections and other life-threatening side effects, the process of choosing treatment methods can be used as a reference for future research.

### 4.3. Consequence Theme: Self-Mastering Dialysis Lives

While HD patients are actively involved in hemodialysis lifestyles, they have the problem-solving ability of symptom management. They are able to find happiness and interests, fulfil their expectations toward life, express a positive perspective on hemodialysis (e.g., treat the dialysis room as work; the dialysis room is the only place you can sleep and relax during work in the world), appreciate when the disease results in benefits (e.g., health is more important than making money or school grade), and experience wellness and good family relationships.

In this study, the four interactive themes exhibited by the self-participating hemodialysis patients emphasized the attainment of results in relation to self-awareness, self-efficacy, and self-control. The patients actively participated in individualized learning to acquire the hemodialysis skills they needed to utilize social resources, regain control over their lives, and understand how they would be able to achieve and appreciate the final outcome related to self-participation in hemodialysis. Throughout this process, the core category was “integrating hemodialysis into my life journey”.

The present study reached data saturation with a sample of 32 HD patients. Since the sample consisted of only well-adapted HD patients from northern Taiwan who had engaged in self-participation, the results obtained for the experiences of these patients are limited regarding the understanding the experiences of self-participation of unadjusted HD patients, which is a limitation of this study. Therefore, future studies may explore the experience of self-participation among unadjusted HD patients. The results from this qualitative study will serve as a reference for the construction of an HD self-participation scale. We believe that incorporating the findings of the present study will help HD patients acquire self-participation HD skills, such that they are able to build new healthy lives for themselves.

## 5. Conclusions

More complete self-participation in hemodialysis positively influences HD patients, allowing them to acquire skills that help incorporate the restrictions of hemodialysis into their lives. In terms of psychological outlook, the HD patients’ willingness to continue living was main finding of our study. Therefore, this requires that nurses recognize patients with lower-level emotional and psychological needs. Secondly, they must provide their professional knowledge and assistance in order to encourage self-participation and motivation to overcome the various life limitations of hemodialysis. Nurses should encourage patients to express their negative emotions or depressive feelings, which appears to reduce such feelings. Nurses should focus on strengthening patients’ will to live, the joy of life, and the discovery of fun in life to increase their self-confidence and perceptions of self-worth. These measures may allow HD patients to gradually construct their roles and tasks, and to transition to the self-mastery of dialysis in their daily lives.

## Figures and Tables

**Figure 1 healthcare-09-01742-f001:**
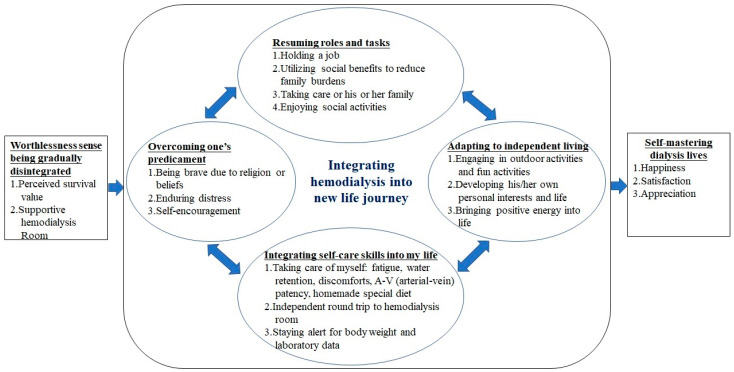
The conceptual framework of integrating hemodialysis into my new life journey.

**Table 1 healthcare-09-01742-t001:** Interview guideline.

Interview Questions
1. What was your impression of your first hemodialysis session? What did you do then? What has changed since then? 2. What did you learn during the hemodialysis experience that helped you to adapt to it? 3. What kind of help did you receive during this process? 4. Can you talk about your self-care preparations for today’s hemodialysis session and post-dialysis period? 5. What type of assistance do you desire most during this process? 6. Based on your experience, what helped you the most to familiarize yourself with hemodialysis treatments? 7. Over what aspects do you feel you have control? What aspects do you not have control of? Why do you feel that way? 8. As someone who has experienced this process, how would you advise a new hemodialysis patient regarding the preparations for commencing self-participation in haemodialysis care?

**Table 2 healthcare-09-01742-t002:** Characteristics of the participants.

Gender	Age (Years)	Education Level	Occupation	Marital Status	HD Duration in Years	Perceived Health Status
F	58	High school	Housewife	Married	20	Fair
F	55	University	None	Widowed	20	Fair
F	55	University	Housewife	Married	20	Fine
M	57	University	None	Married	10	Ordinary
M	45	High school	Service industry	Married	19	Ordinary
F	58	High school	Housewife	Married	25	Ordinary
M	72	Elementary school	Service industry	Married	2.5	Ordinary
M	78	Secondary school	None	Married	1	Ordinary
F	63	Elementary school	None	Married	10	Ordinary
F	64	High school	Housewife	Married	6	Fair
F	84	None	None	Widowed	5	Fair
M	48	College	Part time	Divorce	3	Ordinary
F	85	Secondary school	None	Married	10	Ordinary
F	66	Secondary school	Housewife	Married	13	Ordinary
F	33	University	Worker	Single	18	Ordinary
M	42	University	None	Single	3	Ordinary
M	66	Elementary school	Housewife	Married	6	Ordinary
M	62	College	None	Married	12	Ordinary
F	58	High school	None	Single	3	Ordinary
F	60	Secondary school	Retired	Single	8	Ordinary
M	64	University	Retired	Married	1.5	Ordinary
M	53	University	Chief	Married	3.5	Fair
F	67	Secondary school	Retired	Married	10	Ordinary
M	57	High school	None	Divorce	8	Ordinary
M	63	University	Retired	Married	4	Fine
F	47	University	None	Widowed	3	Ordinary
M	81	Elementary school	None	Widowed	3	Ordinary
F	54	Elementary school	None	Married	8	Ordinary
M	65	High school	Retired	Married	1	Ordinary
F	76	Secondary school	None	Married	3.5	Ordinary
F	72	Secondary school	Housewife	Married	20	Ordinary
M	67	High school	Driver	Married	10	Ordinary

## Data Availability

The datasets used and analyzed during the current study are available from the first author and the corresponding authors on reasonable request.

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
