# Peer review of "Self-Participation Experiences among Well-Adapted Hemodialysis Patients"

_healthcare, 2021, doi:10.3390/healthcare9121742_

Round 1

Reviewer 1 Report

Please see comments in attached pdf file.

Author Response

Please see comments in attached pdf file.

Response: The manuscript has revised according to the comments of the1st reviewer (Rows 13, 15, 16, 18, 26, 60, 61, 75, 88, 93, 100, 114, 154, & 155).

Reviewer 2 Report

The manuscript is easy to read. The topic is also important, it is in line with the message of WKD 2021 “Living Well with Kidney Disease”. The authours focused on hemodialysis patients perspectives using the grounded theory method. The study study included 32 HD patients described as adapted to life as a haemodialysis patient and leading a normal life. They have dialysis vintage of 9.78 years. My major concern is study group selection. I am not sure whether it is the best group to study adaptation to dialysis and plan intervention for the future. I think that adaptation process should be assessed much earlier, no later than after 3-4 years after dialysis start. The participants could have different perspective after surviving of some severe complication related to dialysis. After almost 10 years impression of the first haemodialysis session is completely different than after few weeks, months or one or two years. It would worthy to add information what time is needed for resuming roles and lives and adapting to an independent life. Further, I wonder why patients who were considered not adapted to dialysis were not referred to participate in the study? Is it really real life study?
Moreover, Table 2 is lacking so clinical data are very scarce.  
The interviews were performed during dialysis session instead to be done in a private room. It could have effect on results.
In the Introduction it is stated that  HD patients are unable to undergo normal lives, but later "leading a normal life" is a selection cryterion. Please comment.
There is no information about the number of patients screened for study participation, how big is dialysis facility?
Were study participants on waiting list for kidney transplantation?

Author Response

Point 1:

My major concern is study group selection. I am not sure whether it is the best group to study adaptation to dialysis and plan intervention for the future. I think that adaptation process should be assessed much earlier, no later than after 3-4 years after dialysis start. The participants could have different perspective after surviving of some severe complication related to dialysis. After almost 10 years impression of the first haemodialysis session is completely different than after few weeks, months or one or two years.

Response 1: We recruiting well-adapted participants with diversity of haemodialysis duration in years (range from 1 to 20 years) within the haemodialysis patients according to the theoretical sampling of the Grounded Theory, because they might have different self-participation experience (Ligita, Harvey, Wicking, Nurjannah, & Francis, 2019). Additionally, literature showed that adjustment to ESRD was a dynamic and constant process (Taylor, Taylor, Baharani, Nicholas, & Combes, 2016); HD patients experienced three periods of adjustment (beginning 1-3 weeks from the 1st dialysis of honeymoon period, 3-12 months of frustration period, and the long period of adjustment) (Gerogianni & Babatsikou, 2014); and time on HD was significantly correlated with performance anxiety, sleep disturbance, self-care self-efficacy, health status, mood distress, symptom distress, dialysis stress, and perceived adherence to fluid restriction (Lev & Owen, 1998; Martinez & Custodil, 2014). Therefore, self-participation process and experience can be assessed across HD trajectory.

Point 2: It would worthy to add information what time is needed for resuming roles and lives and adapting to an independent life. Further, I wonder why patients who were considered not adapted to dialysis were not referred to participate in the study? Is it really real life study?

Response 2: The inclusion criteria for participants were already well-adapted to life as a HD patient and leading a normal life. Some studies have showed HD patients unadjusted reactions to HD, such as restricted to a renal world, losing self control, stuck in an endless process (Lin, Han, Pan, 2015); life crisis (Kim & Yang, 2021); facing life’s limitation, living with uncertainty, dependence on medical technology (Chiaranai, 2016); a new dialysis-dependent self, and a restricted life (Reid, Seymour, & Jones, 2016). It might be future study to explore unadjusted HD patients’ participation experience. Literature is lack of self-participation leading to a normal HD life in well-adapted HD patients (Anderen-Hollekim, Solbjor, Kvangarsnes, Hole, & Landstad, 2020). Therefore, our participants were those well-adapted HD patients.

Ligita, T., Harvey, N., Wicking, K., Nurjannah, I., & Francis, K. (2019). A practical example of using theoretical sampling throughout a grounded theory study: A methodological paper. Quantitative Research Journal, 20(1), 116-126. https://doi.org/10.1108/QRJ-07-2019-0059

Taylor, F., Taylor, C., Baharani, J., Nicholas, J., & Combes, G. (2016). Integrating emotional and psychological support into the end-stage renal disease pathway: A protocol for mixed methods research to identify patients’ lower-level support needs and how these can most effectively be addressed. BMC Nephrology, 17: 111. https://doi.org/10.1186/s12882-016-0327-2

Gerogianni, S. K., & Babatsikou, F. P. (2014). Psychological aspects in chronic renal failure. Health Science Journal, 8(2), 205-214.

Lev, e. l., & Owen, S. V. (1998). A prospective study of adjustment to hemodialysis. ANNA J, 25(5), 495-504.

Martinez, B. B., & Custodil, R. P. (2014). Relationship between mental health and spiritual wellbeing among hemodialysis patients: A correlation study. Sao Paulo Med J, 132(1), 23-27. https://doi.org/10.1590/1516-3180.2014.1321606

Point 3: Moreover, Table 2 is lacking so clinical data are very scarce.

Response 3: The Table 2 has added to the results (Rows 116).

Point 4: The interviews were performed during dialysis session instead to be done in a private room. It could have effect on results.

Response 4: The participants choose to be interviewed during dialysis session in a private room. The sentence has been revised (Rows 82-83). Anderen-Hollekim, Solbjor, Kvangarsnes, Hole, & Landstad, (2020) also respect HD patient’s wish to be interviewed during dialysis session.

Point 5: In the Introduction it is stated that HD patients are unable to undergo normal lives, but later "leading a normal life" is a selection criterion. Please comment.

Response 5: In the introduction we stated that the discomfort and impact of HD patients and unable to undergo normal lives. Yet, the inclusion criteria for participants were already adapted to life as a HD patient and leading a normal life. We disclose how do HD patients overcome negative experience and return to normal life. Therefore, our focus is on those HD participant well adapted to HD life and returning a normal life. However, the sentence has revised (Rows 44-45).

Point 6: There is no information about the number of patients screened for study participation, how big is dialysis facility?

Response 6: The information about the number of patients screened for study participation and beds of dialysis facility has added to the manuscript (Rows 76-77 & 113).

Point 7: Were study participants on waiting list for kidney transplantation?

Response 7: We added this information in the results (Row 115).

Reviewer 3 Report

Thank you for giving me the opportunity to review your work. Although it is an interest study, I believe that some modifications are necessary for improving the Quality of your paper. 

  1. Introduction can be enriched with more studies on CKD patients' experience from rendered care. Relevant qualitative studies are published which might be helpful to include in your introductory section.

Providing an operational definition might be also helpful.

Some paragraphs (rows 48-54) must be supported by appropriate references. For example you mention that  previous studies have mostly focused on empowerment and self-management, as well as professional perspectives of patient participation, without reference to those studies. 

  1. Methodology needs to be very detailed. 
    Authors should refer to issues like how the study was advertised, who conducted the interviews, which was the interviewer's experience, how researcher bias was controlled (please use the COREQUE tool for reporting qualitative research). 

In row 82 you mention that all participants chose to be interviewed during a dialysis session. Was this a limitation in your study? how this impact on interview process and on your findings?  

In rows 97 -98 you say that verbatim transcripts of interviews and field notes were analyzed using the method of qualitative interpretive description. Do you mean content analysis? Please provide an explanation on the process of analysis used. 

  1. Results should be presented in a more detailed manner. Ideally there should be representative quotations from all the participants. I understand that you have quite a big sample size, but I think it would be useful if you include some more quotations of your participants in your findings. For example, a short sentence /quotation to highlight each of the distinct issues (sub-themes) that follow the main themes would be beneficial. 

Please use the same names (titles) for themes  throughout the text

  1. Discussion: A more clear presentation of themes is required. 
    It is not clear what exactly the authors mean by the term three antecedent themes (row 188) and how these are presented? 
    “Sense of worthlessness,” “Life is still worth living,” and “Friendly and joyful atmosphere of the haemodialysis room;” are not presented in figure 1. 
    Then we have 4 interactive themes but in fact we have five (row 189) four interactive themes: “Overcoming one’s predicament,” “Integrating self-care skills into my life,” “Resuming previous roles and tasks,” and “Adapting to independent living;” and finally, “Self-mastering haemodialysis lives.” 
    These are a bit confusing for the reader - it is necessary the authors to present in a clearer manner which are the themes /subthemes (or categories /subcategories which is more appropriate term for grounded theory) and how these derived from data. 

Row 263 - Qualitative studies do not aim at generalization, but they focus on studying in depth the unique nature of a phenomenon. So, the authors should highlight this in their paper, since inherently qualitative research does not intend to generalization. 

Grounded theory leads to theory formulation. Which theory derived from this study? 
If your theory has to do with the core category guiding the entire self-participation experience of hemodialysis patients. “Integrating hemodialysis into new life journey” then this is the man finding of your study and need to be adequately highlighted and discussed.  

I do hope these suggestions to assist in improving your work. 

Sincerely yours

Author Response

Response to reviewers’ comments

 Reviewer 3

 Open Review

English language and style

(x) Extensive editing of English language and style required
( ) Moderate English changes required
( ) English language and style are fine/minor spell check required
( ) I don't feel qualified to judge about the English language and style

Yes

Can be improved

Must be improved

Not applicable

Does the introduction provide sufficient background and include all relevant references?

( )

( )

(x)

( )

Is the research design appropriate?

( )

( )

(x)

( )

Are the methods adequately described?

( )

( )

(x)

( )

Are the results clearly presented?

( )

( )

(x)

( )

Are the conclusions supported by the results?

( )

(x)

( )

( )

Comments and Suggestions for Authors

Thank you for giving me the opportunity to review your work. Although it is an interest study, I believe that some modifications are necessary for improving the Quality of your paper. 

  1. Introduction can be enriched with more studies on CKD patients' experience from rendered care. Relevant qualitative studies are published which might be helpful to include in your introductory section.

Providing an operational definition might be also helpful.

Some paragraphs (rows 48-54) must be supported by appropriate references. For example, you mention that  previous studies have mostly focused on empowerment and self-management, as well as professional perspectives of patient participation, without reference to those studies. 

  1. Methodology needs to be very detailed. 
    Authors should refer to issues like how the study was advertised, who conducted the interviews, which was the interviewer's experience, how researcher bias was controlled (please use the COREQUE tool for reporting qualitative research). 

In row 82 you mention that all participants chose to be interviewed during a dialysis session. Was this a limitation in your study? how this impact the interview process and your findings?  

In rows 97 -98 you say that verbatim transcripts of interviews and field notes were analyzed using the method of qualitative interpretive description. Do you mean content analysis? Please explain the process of analysis used. 

  1. Results should be presented in a more detailed manner. Ideally, there should be representative quotations from all the participants. I understand that you have quite a big sample size, but I think it would be useful if you include some more quotations of your participants in your findings. For example, a short sentence /quotation to highlight each of the distinct issues (sub-themes) that follow the main themes would be beneficial. 

Please use the same names (titles) for themes  throughout the text

  1. Discussion: A more clear presentation of themes is required. 
    It is not clear what exactly the authors mean by the term three antecedent themes (row 188) and how these are presented? 
    “Sense of worthlessness,” “Life is still worth living,” and “Friendly and joyful atmosphere of the haemodialysis room;” is not presented in figure 1. 
    Then we have 4 interactive themes but in fact, we have five (row 189) four interactive themes: “Overcoming one’s predicament,” “Integrating self-care skills into my life,” “Resuming previous roles and tasks,” and “Adapting to independent living;” and finally, “Self-mastering haemodialysis lives.” 
    These are a bit confusing for the reader - it is necessary for the authors to present in a clearer manner which are the themes /subthemes (or categories /subcategories which is the more appropriate term for grounded theory) and how these derived from data. 

Row 263 - Qualitative studies do not aim at generalization, but they focus on studying in-depth the unique nature of a phenomenon. So, the authors should highlight this in their paper, since inherently qualitative research does not intend to generalize. 

Grounded theory leads to theory formulation. Which theory was derived from this study? 
If your theory has to do with the core category guiding the entire self-participation experience of hemodialysis patients. “Integrating hemodialysis into new life journey” then is the man finding of your study and need to be adequately highlighted and discussed.  

I do hope these suggestions assist in improving your work. 

Sincerely yours

Submission Date

09 November 2021

Date of this review

26 Nov 2021 21:46:23

Thank you for allowing me to review your work. Although it is an interesting study, I believe that some modifications are necessary for improving the Quality of your paper.

Point 1:

Introduction can be enriched with more studies on CKD patients' experience from rendered care. Relevant qualitative studies are published which might be helpful to include in your introductory section.

Response 1: The introduction has been revised (Rows 37-41).

Point 2: Providing an operational definition might be also helpful.

Response 2: The definition of patient participation has been added to the introduction (Rows 39-41).

Point 3: Some paragraphs (rows 48-54) must be supported by appropriate references. For example, you mention that previous studies have mostly focused on empowerment and self-management, as well as professional perspectives of patient participation, without reference to those studies.

Response 3: The citations of this section have been added (Rows 57, 59, 60-64).

Point 4: Methodology needs to be very detailed.

Authors should refer to issues like how the study was advertised, who conducted the interviews, which was the interviewer's experience, how researcher bias was controlled (please use the COREQUE tool for reporting qualitative research).

Response 4: Well-adapted HD participants were referred by nephrologists and the head nurse of the HD clinic. The section on materials and methods has been revised (Rows 75, 79-80, 81-109).

Point 5: In row 82 you mention that all participants chose to be interviewed during a dialysis session. Was this a limitation in your study? how this impacts the interview process and your findings?

Response 5: We respect participants’ choice of interview time and location. Therefore, the participants choose to be interviewed during dialysis sessions in a private room for saving their time and reserving their energy (Rows 88-89). The sentence has been revised (Rows 82-83). Anderen-Hollekim, Solbjor, Kvangarsnes, Hole, & Landstad, (2020) also respect HD patients’ wish to be interviewed during dialysis session.

Point 6: In rows 97-98 you say that verbatim transcripts of interviews and field notes were analyzed using the method of qualitative interpretive description. Do you mean content analysis? Please explain the process of analysis used.

Response 6: When analyzing data, the constant comparative method is applied to repeatedly compare the existed categories and content. there were several steps during the coding process. This section has been revised (Rows 106-114).

Point 7: Results should be presented in a more detailed manner. Ideally, there should be representative quotations from all the participants. I understand that you have quite a big sample size, but I think it would be useful if you include some more quotations of your participants in your findings. For example, a short sentence /quotation to highlight each of the distinct issues (sub-themes) that follow the main themes would be beneficial.

Response 7: The representative quotations of antecedent themes, interactive themes, and consequence themes have been added and revised (Rows 135-237).

Point 8: Please use the same names (titles) for themes throughout the text.

Response: The themes have been revised for using the same name throughout the text (Rows 135-223).

Point 9: Discussion: A more clear presentation of themes is required.
It is not clear what exactly the authors mean by the term three antecedent themes (row 188) and how these are presented?
“Sense of worthlessness,” “Life is still worth living,” and “Friendly and joyful atmosphere of the haemodialysis room;” is not presented in figure 1.T
Then we have 4 interactive themes but in fact, we have five (row 189) four interactive themes: “Overcoming one’s predicament,” “Integrating self-care skills into my life,” “Resuming previous roles and tasks,” and “Adapting to independent living;” and finally, “Self-mastering haemodialysis lives.”
These are a bit confusing for the reader -it is necessary for the authors to present in a clearer manner which are the themes /subthemes (or categories /subcategories which is more appropriate term for grounded theory) and how these derived from data.

Response 8: The first paragraph of the discussion has revised (Row 240-244).

Point 9: Row 263 Qualitative studies do not aim at generalization, but they focus on studying in depth the unique nature of a phenomenon. So, the authors should highlight this in their paper, since inherently qualitative research does not intend to generalization.

Grounded theory leads to theory formulation. Which theory was derived from this study?
If your theory has to do with the core category guiding the entire self-participation experience of hemodialysis patients. “Integrating hemodialysis into new life journey” then is the main finding of your study and needs to be adequately highlighted and discussed.

Response 9: The discussion of the consequence theme and limitation has been revised (Row 318-331).

Round 2

Reviewer 2 Report

The authors presented the corrected version of their manuscript focused on patients’ perspectives on  integration of hemodialysis into a new life. Answers to my comments were adequately addressed but not included in the text (mainly point 1). I suggest to add it to introduction or discussion. It would be worthy to add some more raw data, the most frequent answers to questions. Were there any difference between very long HD survivors and patients with shorter HD vintage? Does 126 dialysis beds mean that you have 500-600 patients? Can your results be generalised? Please explain why some patients were not transplanted but dialyzed several years. 

Author Response

The authors presented the corrected version of their manuscript focused on patients’ perspectives on integration of hemodialysis into a new life. Answers to my comments were adequately addressed but not included in the text (mainly point 1). I suggest to add it to introduction or discussion. It would be worthy to add some more raw data, the most frequent answers to questions. Were there any difference between very long HD survivors and patients with shorter HD vintage?

Response: It was revised at discussion as suggested. The paragraph of the discussion has revised (Row 271-276, 291-295, 348-350) and conclusions (389-392).

Does 126 dialysis beds mean that you have 500-600 patients? Can your results be generalized?

Response: Yes, 126 dialysis beds mean that you have 500-600 patients. However, in the current study recruited the well-adapted HD patients by using theoretical sampling to focus on the initial self-participation experiences. It was revised in the recruiting criteria, (d) they have identified themselves as well-adapted HD, although some interviewed HD patients were very long HD survivors who still have a vivid memory of initial self-participation experiences and were willing to share these experiences. The result provided a substantive descriptive theory based on grounded theory instead of generalization of all HD populations.

It was revised at material and methods as suggested (Rows 89-91).

Were there any difference between very long HD survivors and patients with shorter HD vintage?

Response: Shorter HD patients have psychological pressure on the treatment of twice insertion (centrifugation or centripetal) pain and A-V shunt dysfunction than long-term HD patients

Please explain why some patients were not transplanted but dialyzed several years.

Response: This is not the purpose of the current study. These patients did not mention that they are waiting for a kidney transplant. They have adapted to the current dialysis life. considering the complications of kidney transplantation, such as infections and rejections, and other life-threatening side effects, the process of choosing treatment methods Can be used as a reference for future research.

It was revised at material and methods as suggested (Rows 352-355).

Reviewer 3 Report

Dear Authors 

your paper has been sufficiently improved

please omit parentheses from WHO (lines 39, 41)

Best wishes  

Author Response

please omit parentheses from WHO (lines 39, 41)

Response: Thank you for giving me the remind.